# Long-Term Serological Investigations of Influenza A Virus in Free-Living Wild Boars (*Sus scrofa*) from Northern Italy (2007–2014)

**DOI:** 10.3390/microorganisms10091768

**Published:** 2022-09-01

**Authors:** Maria Alessandra De Marco, Claudia Cotti, Elisabetta Raffini, Matteo Frasnelli, Alice Prosperi, Irene Zanni, Chiara Romanini, Maria Rita Castrucci, Chiara Chiapponi, Mauro Delogu

**Affiliations:** 1Institute for Environmental Protection and Research (ISPRA), 40064 Ozzano dell’Emilia, Italy; 2Wildlife and Exotic Animal Service, Department of Veterinary Medical Sciences, University of Bologna, 40064 Ozzano dell’Emilia, Italy; 3WOAH Reference Laboratory for Swine Influenza, Istituto Zooprofilattico Sperimentale della Lombardia e dell’Emilia Romagna (IZSLER), 25124 Brescia, Italy; 4Department of Food Safety, Nutrition and Veterinary Public Health, Istituto Superiore di Sanità, 00161 Rome, Italy; 5Biochemistry and Molecular Biology Unit, Department of Life Sciences, University of Parma, 43124 Parma, Italy

**Keywords:** influenza A virus, ecology, *Sus scrofa*, wild boar, Italy, antibodies, enzyme-linked immunosorbent assay, hemagglutination inhibition assay, serological investigation

## Abstract

Influenza A viruses (IAV) have been repeatedly demonstrated to circulate in wild suid populations. In this study, serum samples were collected from 2618 free-ranging wild boars in a protected area of Northern Italy between 2007 and 2014, and firstly screened by enzyme-linked immunosorbent assay (ELISA) for the presence of antibodies against IAV. The ELISA-positive samples were further tested by hemagglutination inhibition (HI) assays performed using antigen strains representative of the four major swine IAV (sIAV) lineages circulating in Italy: avian-like swine H1N1, pandemic-like swine H1N1, human-like swine H1N2 and human-like swine H3N2. An overall seroprevalence of 5.5% (145/2618) was detected by ELISA, with 56.7% (80/141) of screened sera tests positive by HI assay. Antibodies against H1N1 subtypes were the most prevalent beginning in 2009—with the highest detection in the first quarter of the year—until 2013, although at a low level. In addition, antibodies to H3N2 subtype were found during six years (2007, 2009, 2010, 2011, 2012 and 2014) whereas H1N2 antibodies were detected in 2012 only. Of the HI-positive samples, 30% showed reactivity to both H1N1 and H3N2 subtypes. These results provide additional insight into the circulation dynamics of IAV in wild suid populations, suggesting the occurrence of sIAV spillover events from pigs to wild boars.

## 1. Introduction

Influenza A viruses (IAV) can infect a large variety of animal species, including humans, pigs, horses, sea mammals and birds, and ecological interactions occurring worldwide among susceptible hosts enhance the virus’s ability for cross-species transmission [1]. Scientific evidence shows that the aquatic birds of the world enable the perpetuation of IAV, classified to date into 16 haemagglutinin (H1–H16) and 9 neuraminidase (N1–N9) antigenic subtypes of avian influenza viruses (AIV) [2]. Occasional transmissions of IAV from the natural avian reservoir to different susceptible species of birds and mammals can lead to variable degrees of virus adaptation in colonized hosts, ranging from sporadic cases to the sustained circulation of species-adapted IAV lineages, for example, in poultry, humans, pigs, horses and dogs [2].

Pigs harbor established lineages of swine influenza viruses (sIAV) and, due to the expression of dual receptors for avian and mammalian IAV, enable the adaptation of AIV to mammals as well as the reassortment of IAV genes of different species, representing a perfect “mixing vessel” [3,4] involved in the generation of influenza viruses potentially pandemic for humans [5,6]. A number of AIV, including H3N3, H4N6, H5N2 and H9N2 antigenic subtypes, have been isolated from swine [2], whereas established lineages of sIAV belonging to the H1N1, H1N2 and H3N2 subtypes and their reassortants have been repeatedly reported in pig populations reared worldwide [7,8]. Due to the evolutionary complexity of sIAVs, recently, the H1 genetic sublineages were classified according to an alphanumerical nomenclature [9]. In the context of ecological interactions taking place at this human–animal interface, the frequent bidirectional transmission of host-adapted IAV occurring between pigs and humans has allowed the human pandemic A/H1N1 2009 (H1N1pdm) virus to infect pig herds and to generate novel reassortants with co-circulating lineages of sIAV [10,11].

Interestingly, from an ecological perspective, domesticated pigs (*Sus scrofa domesticus*) and their wild ancestor, the wild boar (*Sus scrofa*), coexist and share interbreeding potential [12] as well as susceptibility to most pathogens, including IAV [13]. Epidemiological studies have repeatedly demonstrated the circulation of IAV and/or sIAV in Eurasian populations of wild boar [13,14,15,16,17,18,19,20,21,22,23,24,25,26,27,28,29], showing variable seroprevalences ranging from 1.1% to 24.2% [13,21] and virological prevalences from 0.4% to 3.4% [15,19,29]. Indirect and direct evidence of wild suid exposure to the A/H1N1 2009 (H1N1pdm) virus (including its reassortants) has also been described [20,28,29,30]. It is noteworthy that, after the previous lack of evidence of AIV transmission to wild boars [31] antibodies to H5N8 highly pathogenic avian influenza (HPAI) viruses have been reported in wild boars in Germany [28].

Only a few studies are available about the presence in Italy of IAV infection in free-living wild boars, all reported in the northern region of Emilia Romagna [19,27,29] in the Po Valley, which is the main representative area for swine production (63% of Italian pig production, with 5,271,455 of 8,407,968 animals as of December 2021) [32]. Four major sIAV lineages circulate in Eurasian [8] and Italian pigs [33,34]: avian-like swine H1N1 (H1N1av), pandemic-like swine H1N1 (H1N1pdm), human-like reassortant swine H1N2 (H1huN2) and human-like reassortant swine H3N2.

To better understand the epidemiological role played by free-living wild boar populations in the circulation dynamics of these sIAV subtypes, we have carried out an eight-year serological investigation in a free-ranging wild boar population under selective control in the Gessi Bolognesi e Calanchi dell’Abbadessa Regional Park, an anthropized protected area of the Emilia–Romagna region of Northern Italy.

## 2. Materials and Methods

### 2.1. Study Area

The Regional Park of Gessi Bolognesi e Calanchi dell’Abbadessa (park headquarters 44°25′52.1″ N, 11°24′08.0″ E) is a Natura 2000 site (Sites of Community Importance–Special Protection Area, SCI–SPA IT4050001) located in a hilly area within the municipalities of Bologna, Ozzano dell’Emilia, Pianoro and San Lazzaro di Savena in the Emilia Romagna Region of Northern Italy (with some altitudes ranging from 64 to 375 m a.s.l.). Over 10,000 residents live in this unfenced protected area, which covers 4844 hectares characterized by natural chalk outcrops and erosion furrows, and including wooded, bush and grass zones as well as a large area of usable agricultural land (Figure 1) [35]. A wild boar population management and control plan is implemented in the study area [36].

### 2.2. Sample Collection

During the management and control plan implemented in the Gessi Bolognesi e Calanchi dell’Abbadessa Regional Park, between 20 February 2007 and 30 September 2014, serum samples were collected from wild boars trapped and shot (*n* = 2457) or released into the wild (*n* = 161) according to authorized activities carried out to preserve animal welfare.

In addition, 27 of the released wild boars, which had been previously captured by cage traps and individually identified by ear-tag number, were recaptured one or more times at least 2 weeks after the previous sampling, allowing us to assess possible seroconversions.

Blood clots collected from wild boars were kept at 4 °C and centrifuged within 24 h of collection, and sera were stored at −20 °C until tested.

During sampling, wild boars were sexed and aged (in months) according to morphology and tooth development [37] as follows: 0 mo. < age class 1 ≤ 6 mo.; 6 mo. < age class 2 ≤ 14 mo.; 14 mo. < age class 3 ≤ 24 mo.; age class 4 > 24 mo. Moreover, for interpreting young wild boar seroreactivity, possibly related to maternal passive immunity (usually disappearing in pigs by age 4–14 weeks) [38,39] or elicited by active serologic response, class 1 was further categorized into the following subclasses: 0 mo. < 1st subclass 1 ≤ 3 mo., and 3 mo. < 2nd subclass 1 ≤ 6 mo.

### 2.3. Serological Methods

Serum samples were screened for the presence of anti-nucleoprotein (NP) antibodies using a validated homemade ELISA test (NP-ELISA) [19,40] with a sensitivity of 94.5% (95% confidence interval [CI] 91.6–96.7) and a specificity of 99.2% (95% CI 98.4–99.7) performed according to de Boer et al. [41], and positive sera were further tested by hemagglutination inhibition (HI) assay [42]. In brief, sera were treated with potassium periodate-trypsin as described previously [43] and adsorbed onto turkey erythrocytes. HI tests were performed using four hemagglutinating units of virus and 0.5% turkey erythrocytes. The HI assay cut-off was set to 1:20 [19]. Sera were examined against IAV strains representative of the following H1N1, H1N2, H3N2 antigenic subtypes circulating in Italian pig farms in the late 1990s and in early 2000s: A/swine/Italy/1513/1998/H1N1av (hereinafter named H1N1/98), A/swine/Italy/1521/1998/H1huN2 (named H1N2/98), A/swine/Italy/1523/1998/H3N2 (named H3N2/98), A/swine/Italy/311368/2013/H1N1av (named H1N1/13), A/swine/Italy/282866/2013/H1N1pdm (named H1N1pdm/13), A/swine/Italy/284922/2009/H1huN2 (named H1N2/09), and A/swine/Italy/311349/2013/H3N2 (named H3N2/13).

In order to assess possible seroconversions in the recaptured wild boars, detection of antibodies in previously negative samples or a minimum four-fold increase in titer were considered suggestive of IAV infection occurring during the study period.

### 2.4. Statistical Methods

Fisher’s exact test and chi-square test were performed to test the wild boar population for statistically significant differences in IAV seroprevalences when comparing sampling years, and sexes and age classes/subclasses within each year (EPISTAT 3.3, Epistat Services, Richardson, TX, USA). For both tests, the significance threshold was set at a *p*-value < 0.05. For the entire study period, geometric means of HI reciprocal titers (GMT) were also calculated for each IAV strain used in the HI assay. In calculating GMT, an HI reciprocal titer < 20 was assigned a value of 10.

### 2.5. Animal Rights Statement

No specific ethical approval was required for this study, as biological samples were collected from free-living wild boars trapped and/or shot by wildlife control operators in compliance with the Italian Protected Areas Law 394/1991 and Italian Hunting Law 157/92. In particular, sampling activities were scheduled in the Wild Boar Management and Control Plan implemented in the protected area Gessi Bolognesi e Calanchi dell’Abbadessa Regional Park in the 2005–2009 and 2010–2014 periods according to Park Executive Committee deliberations upon technical evaluation of ISPRA (Italian Institute for Environmental Protection and Research). During this study, all wild boars were handled and sampled in accordance with standardized procedures [44].

## 3. Results

### 3.1. Study Populations

As inferred from Table 1, 29.7%, 24.4%, 20.7%, 16.5% and 8.6% of the 2618 wild boars under study—consisting of 1233 females, 1239 males and 146 unsexed animals—grouped, respectively, into age classes 1, 2, 3, 4 and unspecified.

The number of sampled animals ranged from a minimum value of 76 in 2008 to a maximum value of 480 in 2011 during the eight-year study period, with at least 9 months sampling for each year (Figure A1). Twenty-seven wild boars were also recaptured one or more times.

### 3.2. Serological Results

In the present study, 145/2618 captured wild boars (Table 1, Figure 2) and 3/27 recaptured wild boars tested IAV seropositive by NP-ELISA. Due to the limited volume of 7 NP-ELISA screened samples (Appendix A), only 141 wild boars (including 138 captured and 3 recaptured animals) were further analyzed by HI assay (Table 2 and Appendix A).

Overall, 5.5% of 2618 wild boars tested positive for IAV, with the lowest (0.6%) and highest (17.6%) NP-ELISA seroprevalence values during 2013 and 2009, respectively. No IAV antibodies were detected in the 76 samples collected in 2008 (Table 1, Figure 2a). Significantly higher IAV seroprevalence values were found in 2007 vs. 2008 and 2010–2013; in 2009 vs. 2007, 2008 and 2010–2014; in 2010 vs. 2013 and 2014; and in 2012 vs. 2011, 2013 and 2014.

No sex-related differences in proportions of IAV seropositive wild boars were detected during any year. With regard to age-related differences, in 2007, the 1st subclass 1 showed significantly higher IAV seroprevalence when compared with the 2nd subclass 1, class 2, and class 3 (Table 1), whereas in 2009, this 1st subclass showed significantly lower seroprevalence than the remaining classes 2, 3 and 4. Significantly higher seroprevalence was also detected in the 2nd subclass 1 vs. class 2 in 2012; in class 3 vs. class 2 in 2011 and 2012; and in class 4 vs. 2nd subclass 1 in 2007.

For the 2012 samples, the NP-ELISA results included in the present study were previously obtained by Delogu et al. [27], whereas HI assays were performed again using the viruses reported below. In particular, HI-positive results were obtained from 80 out of 141 available IAV positive sera, using as antigens seven IAV strains representative of H1N1, H1N2 and H3N2 antigenic subtypes circulating in Italian pig farms in the late 1990s and in the first two decades of the 2000s. Specifically, HI seroreactivity to the following antigenic subtypes of swine isolates of influenza viruses were detected: H1N1 in 2009, 2010, 2011, 2012 and 2013 (45/47, 8/8, 4/8, 1/18 and 1/2, respectively); H1N2 in 2012 (4/18); H3N2 in 2007, 2009, 2010, 2011, 2012 and 2014 (10/51, 21/47, 2/8, 2/8, 2/18 and 1/4, respectively) (Figure 2b). Overall, 56.7% of sera tested IAV-positive were found to be HI-positive for at least one sIAV strain; however, variable percentages were observed during the entire study period, ranging from 100% (e.g., 15/15 and 12/12 in March and April 2009, respectively) to 7.7% (e.g., 1/13 in May 2007) (Figure A1). This discrepancy between NP-ELISA and HI results, as has been previously described [19,27,28], could be related to different types of antibodies targeting either the conserved NP or the variable hemagglutinin of IAV, and possibly differing in development and decline [41]. In addition, sIAV not represented in our panel of HI antigens could have accounted for seropositivity detected only by the NP-ELISA and not by HI assay [19,27,28]. Finally, the high sensitivity of NP-ELISA (94.5%) used to screen a large set of serum samples at a threshold dilution of 1:10 [40] could also have accounted for the lack of detection of low antibody levels by HI assay (1:20 threshold dilution) [42].

As detailed in Table 2 and Appendix A, the seven sIAV strains included: two Eurasian avian-like swine H1N1/98 (lineage 1C.1-2-like) and H1N1/13 (lineage 1C.2.1), and pandemic-like swine H1N1pdm/13 (lineage 1A.3.3.2); two human-like reassortant swine H1N2/98 (lineage 1B.1.2.2) and the H1N2/09 (lineage 1B.1.2.2); and two human-like reassortant H3N2/98 and H3N2/13.

Taken together, the H1N1 HI-positive results detected beginning in February 2009—with the highest detection in the first quarter of the year—until March 2013 (Table 2, Figure A1) showed a total (including seropositive recaptured wild boars) of 62, 61 and 58 wild boars seropositive for the H1N1/98, H1N1/13 and H1N1pdm/13 strains, respectively, sequentially showing overall GMT (antibody level range) of 26.1 (20–1280), 22.0 (20–640) and 18.2 (20–320). In particular, high HI reactivity was observed for the two Eurasian avian-like H1N1 viruses, with a four-fold titer difference in three samples only (ID 2754, ID 2894 and ID 3044). However, serum samples that tested positive for Eurasian avian-like H1N1 strains were also reactive to the H1N1pdm-like virus at a slightly lower level, except for samples ID 2608, ID 2612, ID 2628 and ID 3376, which showed ≥ four-fold differences in lower titers. Although the cross-reactivity between these antigens might influence the interpretation of results [28,45,46], the high HI seroreactivity of Eurasian avian-like H1N1 strains detected in early 2009 might impact possible subsequent infections with H1N1pdm-like viruses [47].

Compared to results obtained with older and more recent antigen strains, both the recent H1N2 and H3N2 lineages revealed higher HI titers, appearing to be more efficient at detecting antibody levels in seropositive wild boars. With regard to the H1N2 strains, only 4 of 18 IAV seropositive wild boars sampled in 2012 were HI-positive for the H1N2/09 strain between 21 June and 5 October, with overall GMT (antibody level range) of 10.2 (20–40). Finally, considering the H3N2 HI-positive results detected from 9 March 2007 to 12 June 2014, a total of 3 and 39 wild boars were seropositive for the H3N2/98 and H3N2/13 strains, respectively, with overall GMT (antibody level range) of 10.1 (20–20) and 13.0 (20–80).

With regard to individual multiple-seropositivity to different subtypes, 24 wild boars tested HI-positive for both H1N1 and H3N2 subtypes, whereas only one wild boar was seropositive against H1N2 and H3N2 strains (Table 2).

Twenty-seven wild boars, individually identified by ear-tag number, were recaptured one or more times at least 2 weeks after the previous sampling, allowing us to assess seroconversions in three wild boars. The first one was found to be IAV seronegative on 1 July 2008 and tested H1N1 and H3N2 positive on 10 March 2009 (male of class 2, ID 2623). The other two wild boars were IAV seronegative on 4 June 2008 and 18 July 2007 but showed seroconversions to H1N1 viruses on 7 May 2010 (male of class 3, ID 2940) and 18 May 2010 (male of class 4, ID 2955) (Table 2). The remaining 24 wild boars recaptured one (*n* = 18) or two (*n* = 6) times between 15 May 2007 and 14 September 2014 always tested IAV seronegative.

## 4. Discussion

Their high prolificacy and resilience to multiple environmental and climatic contexts combined with the reforestation of hilly/mountainous environments and indiscriminate release actions underlie the wild boar demographic expansion in Italy. The consequent impacts to the environment, biodiversity and human activities are normally mitigated by targeted control strategies, including the removal of individuals (capture and/or selective shooting) [37].

This study, carried out during the Wild Boar Management and Control Plan implemented in the protected area Gessi Bolognesi e Calanchi dell’Abbadessa Regional Park, was designed to evaluate eco–epidemiological aspects of IAV infection in a free-living *Sus scrofa* population found in 2012 to be IAV seropositive and variably reactive to sIAV subtypes circulating in Italian pig farms [27]. Indeed, four major sIAV lineages were circulating in Italian pigs: H1N1av, H1N1pdm, H1huN2 and H3N2 [48,49], and the possibility that pigs are a source of sIAV for wild boars has been hypothesized [19,27,29].

From an ecologic perspective, domesticated pigs (*Sus scrofa domesticus*) and wild boars (*Sus scrofa domesticus*) share interbreeding potential and susceptibility to most pathogens, including sIAVs [20], and the Gessi Bolognesi e Calanchi della Abbadessa Park study area may provide wildlife–livestock ecological interfaces, enabling sIAV spillover events. Indeed, some free-range swine farms were present in the Ozzano dell’Emilia, Pianoro and San Lazzaro di Savena municipalities during the study period [50] and clear evidence of pig–wild boar interbreeding was observed in a young individual captured during the control plan implemented in this park (Figure 3).

Moreover, from an epidemiological perspective, this unfenced study area, characterized by a remarkable contiguity between woodland and open cultivated areas (Figure 1), provides refuge and food for these ungulates, whose unrestricted movement and natural aggregation could enhance IAV infection, transmission, spread and endemization.

To better understand the IAV circulation dynamics in wild boars from the Gessi Bolognesi e Calanchi dell’Abbadessa Regional Park area, we screened by NP-ELISA sera collected from captured (n = 2618) and recaptured (n = 27) wild boars between 2007 and 2014, and then, IAV-positive sera were tested by HI assay using seven antigens including sIAV strains circulating in Italy before 2000 and between 2009 and 2013 (Table 2 and Appendix A).

Overall, 5.5% of 2618 wild boars tested positive for IAV, with the lowest (0%) and highest (17.6%) NP-ELISA seroprevalence values during 2008 and 2009, respectively (Table 1, Figure 2a). Antibodies to IAV were found during all years except 2008 (Table 1), when logistical problems in collecting sera led to a very small sample size (n = 76), which might not have permitted the detection of IAV seroprevalence lower than 5% for an estimated population density of wild boars of around 7.2 animals/km^2^ (95% confidence level) [51].

As shown in the Results section, significantly higher seroprevalences between sampling years were found at a variable level. However, annual (Figure 2) and monthly (Figure A1, Table 2) distributions of seropositive wild boars revealed at least three possible estimated peaks of infection based on NP-ELISA and/or HI results, similar to the results shown by Pepin et al. [52] in wild pig populations from 15 U.S. states. In our study, the first estimated peak (from April to July 2007) for an H3N2 infection occurred at the beginning of 2007, the second peak (from February to July 2009) for H1N1 and/or H3N2 infections in late 2008/early 2009, and the third peak (from June to November 2012) for an H1N2 infection in spring/summer 2012. In particular, the second estimated peak was likely related to the recent introduction of the avian-like H1N1 subtype that may have spread in an immunological naive population by infecting almost all age classes (Table 2 and Appendix A). Moreover, antibody detection in three wilds boars (ID 2602, ID 2603 and ID 2604) belonging to the 2nd subclass of age class 1 (without potential maternal immunity) further suggests recent H1N1 virus introduction in this wild boar population. In particular, one of these animals (ID 2604) also tested HI-positive for the H3N2/13 strain (Table 2 and Appendix A).

During the study period, we observed a high H1N1 seroreactivity, followed by a decreasing trend in the available IAV-positive samples, amounting to 95.7% (45/47) in 2009, 100% (8/8) in 2010, 50% (4/8) in 2011, 5.6% (1/18) in 2012, 50% (1/2) in 2013 and finally 0% (0/4) in 2014 (Table 1). As previously stated in the Results section, apart from cross-reactivity between H1N1 subtypes [28,45,47], HI reactivity revealed in the present study likely accounts for antibodies to the avian-like swine H1N1 lineage appearing before the H1N1pdm emergence in pigs in late 2009 [53] and as also reported in European wild boars in Spain [14], Germany [15,18,28], Croatia [17] and Italy [19,27]. With regard to the H3N2 subtypes, we found HI seroreactivity in IAV-positive samples amounting to 19.6% (10/51) in 2007, 44.7% (21/47) in 2009, 25.0% (2/8) in 2010, 25% (2/8) in 2011, 11.1% (2/18) in 2012 and 25% (1/4) in 2014 (Figure 2). In Europe, antibodies to human-like swine H3N2 have been found in wild boars in Germany [15,28]. Finally, H1N2/09 seroreactivity was only found in 2012 (4/18 IAV-positive samples), and detection of antibodies in two wild boars (ID 3903, ID 3904) belonging to the 2nd subclass of age class 1 accounts for a recent infection with this human-like virus (Table 2 and Appendix A). Accordingly, only sporadic evidence of H1N2 seroprevalence was reported in European wild boars [28]. Furthermore, sIAV seroconversion found in three recaptured wild boars in March 2009 and May 2010 corroborates the sustained circulation of H1N1 viruses and the lower circulation of H3N2 viruses in the wild boar population, as revealed by the antibody levels detected in 2009 (Figure 2 and Figure A1).

Overall, seropositive results obtained for avian-like swine H1N1, human-like swine H1N2 and human-like swine H3N2 reflect the periodic introduction of sIAV circulating in pig populations from Northern Italy [48,49]. Notably, recent data showed significant genetic differentiation in sIAV from pigs reared in the Emilia–Romagna region and simultaneous circulation of the same viral genotypes in pigs and wild boars in overlapping years, suggesting the hypothesis of continuous introduction of viral strains of sIAV from domestic to wild suid population [29].

## 5. Conclusions

From an ecological perspective, domesticated pigs (*Sus scrofa domesticus*) and their wild ancestor, the wild boar (*Sus scrofa*), coexist and share interbreeding potential [12] as well as susceptibility to most pathogens, including IAV [13].

Providing new insights into wild boar antibody responses to sIAVs, our findings show that periodic introductions of sIAV may occur in the wild boar population sampled in a protected area. In particular, our serological investigation revealed epidemic trends of H3N2 infection in 2007, H1N1 and/or H3N2 infection in 2009, and H1N2 infection in 2012.

Therefore, serological and virological monitoring of wild boar populations should be implemented to evaluate IAV exposure and to identify spillover events at the livestock–wildlife interface and potential emergence of zoonotic viruses.

## Figures and Tables

**Figure 1 microorganisms-10-01768-f001:**
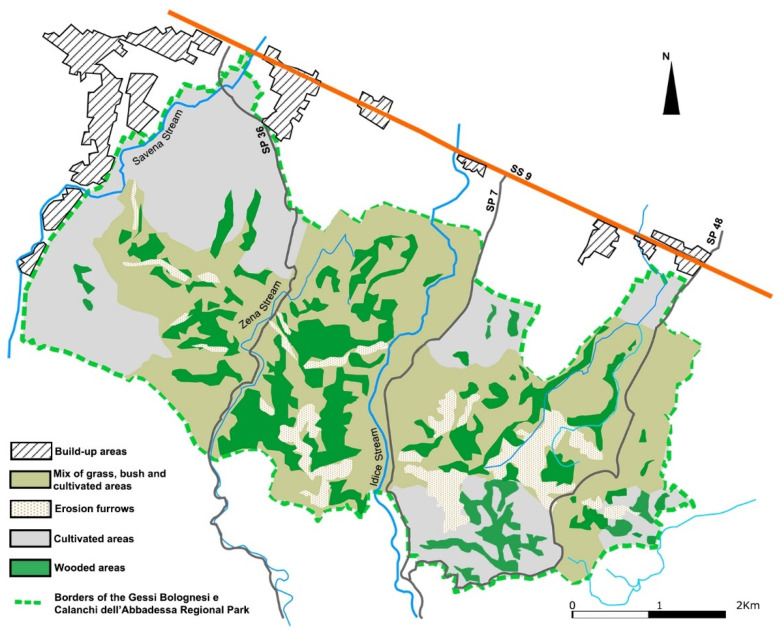
Protected area of Northern Italy where 2645 wild boar (*Sus scrofa*) sera were examined by serological assays to assess influenza A virus circulation between 2007 and 2014.

**Figure 2 microorganisms-10-01768-f002:**
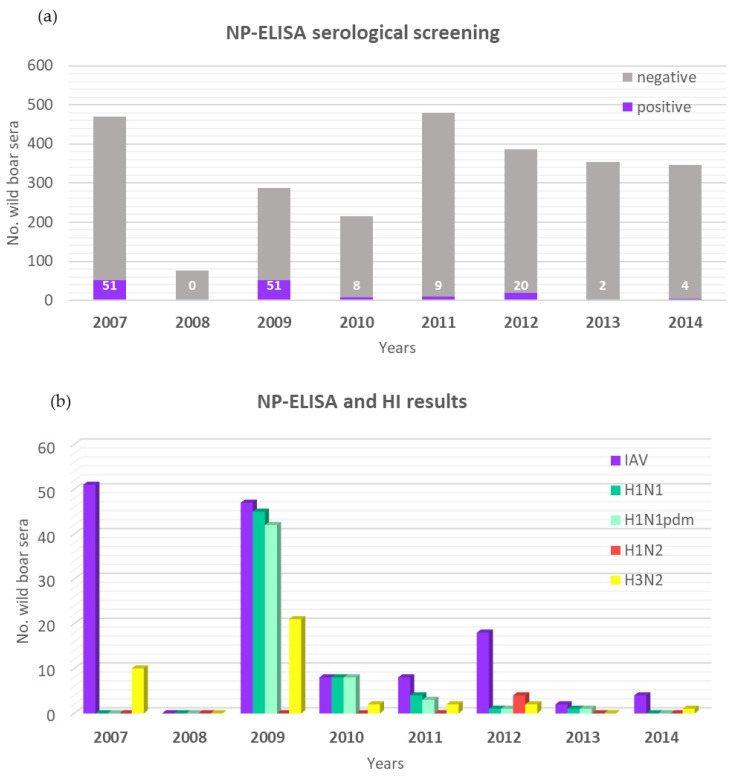
Serological evidence of influenza A virus (IAV) exposure in wild boars (*Sus scrofa*) from an anthropized area of Northern Italy (2007–2014): (**a**) Wild boars screened for the presence of anti-influenza A nucleoprotein (NP) antibodies by NP-ELISA. (**b**) NP-ELISA-positive wild boars examined by HI assay against sIAV subtypes representative of viruses circulating in Italian pig farms. Recaptured wild boars are not included in this figure. See Table 2 for details about the seven sIAV strains used.

**Figure 3 microorganisms-10-01768-f003:**
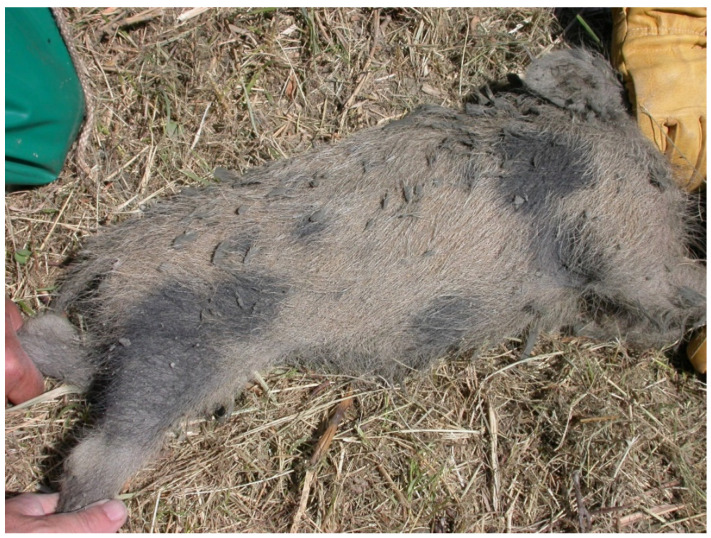
Free-ranging suid captured in the Gessi Bolognesi e Calanchi della Abbadessa Regional Park showing a typical hybrid morphological (spotted) phenotype.

**Table 1 microorganisms-10-01768-t001:** Overall results from wild boars (*Sus scrofa*) tested for the occurrence of influenza A virus exposure in an anthropized area of Northern Italy (2007–2014).

Sampling Year	Sample Size	No. of Seropositive/Wild Boars Tested by NP-ELISA								NP-ELISA Seroprevalence%
		Age Class *					Sex			
		1 ^	2	3	4	U	F	M	U	
2007	469	30/188	5/88	4/62	3/16	9/115	20/175	23/199	8/95	**10.9** (51/469)
2008	76	0/13	0/27	0/13	0/3	0/20	0/32	0/42	0/2	0/76
2009	289	3/67	23/89	16/83	4/27	5/23	22/144	26/135	3/10	**17.6** (51/289)
2010	217	1/82	0/36	5/78	1/17	1/4	5/98	2/112	1/7	**3.7** (8/217)
2011	480	2/160	0/120	5/124	2/67	0/9	6/234	3/238	0/8	1.9 (9/480)
2012	386	13/146	1/119	5/60	1/56	0/5	8/179	12/199	0/8	**5.2** (20/386)
2013	354	0/53	1/88	0/71	1/98	0/44	1/191	1/151	0/12	0.6 (2/354)
2014	347	0/69	0/72	0/52	4/148	0/6	1/180	3/163	0/4	1.1 (4/347)
Total	2618	778	639	543	432	226	1233	1239	146	5.5 (145/2618)

* Age classes: 0 mo. < age class 1 ≤ 6 mo.; 6 mo. < age class 2 ≤ 14 mo.; 14 mo. < age class 3 ≤ 24 mo.; age class 4 > 24 mo.; U, undetermined. ^ for statistical analysis, as shown in Appendix A, age class 1 was further categorized into: 0 mo. < 1st subclass 1 ≤ 3 mo., and 3 mo. < 2nd subclass 1 ≤ 6 mo. Bolded overall seroprevalence values indicate a significantly higher percentage using *p*-value of <0.05. See Serological Results section for statistical results details. Recaptured wild boars are not included in this table.

**Table 2 microorganisms-10-01768-t002:** Hemagglutination inhibition assay (HI) reciprocal titers obtained from 141 available wild boar (WB) that had sera tested positive for influenza A virus nucleoprotein (NP) by NP-ELISA (Northern Italy, 2007–2014). The HI assays were performed using the following strains of inactivated influenza A viruses (IAV): H1N1/98, A/swine/Italy/1513/1998/H1N1av; H1N1/13, A/swine/Italy/311368/2013/H1N1av; H1N1pdm/13, A/swine/Italy/282866/2013/H1N1pdm; H1N2/98, A/swine/Italy/1521/1998/H1huN2; H1N2/2009, A/swine/Italy/284922/2009/H1huN2; H3N2/98, A/swine/Italy/1523/1998/H3N2; and H3N2/13, A/swine/Italy/311349/2013/H3N2.

Sampling Date (No./IAV pos./HI pos.)				HI Titers from WB-Tested IAV-Positive by NP-ELISA						
	WB Serum ID	Age Class *	Sex	H1N1/98	H1N1/13	H1N1pdm/13	H1N2/98	H1N2/09	H3N2/98	H3N2/13
9 March 2007 (1/1/1)	1984	U	U	-	-	-	-	-	-	20
5 April 2007 (1/1/1)	2035	2	F	-	-	-	-	-	-	20
22 May 2007 (11/4/1)	2125	3	M	-	-	-	-	-	-	20
12 June 2007 (12/6/2)	2175	U	U	-	-	-	-	-	-	40
	2180	1	M	-	-	-	-	-	-	20
14 June 2007 (18/14/2)	2187	1	F	-	-	-	-	-	-	20
	2197	U	U	-	-	-	-	-	-	20
26 June 2007 (1/1/1)	2222	1	M	-	-	-	-	-	-	20
3 July 2007 (2/2/2)	2230	1	M	-	-	-	-	-	-	20
	2231	1	F	-	-	-	-	-	-	20
26 February 2009 (7/6/6)	2600	2	M	320	160	80	-	-	-	-
	2601	2	F	160	80	40	-	-	-	-
	2602	1	M	40	80	40	-	-	-	-
	2603	1	F	80	40	40	-	-	-	-
	2604	1	M	40	40	20	-	-	-	20
	2606	3	F	80	80	40	-	-	-	20
27 February 2009 (1/1/1)	2608	2	M	320	320	80	-	-	-	20
4 March 2009 (5/4/4)	2611	2	M	40	40	20	-	-	-	-
	2612	2	M	160	160	40	-	-	-	20
	2613	2	F	80	40	20	-	-	-	20
	2614	2	M	20	20	-	-	-	-	40
10 March 2009 (7/2/2)	2623 (^§^)	2	M	80	40	40	-	-	-	20
	2624	2	F	80	40	40	-	-	20	40
12 March 2009 (3/3/3)	2627	2	M	160	80	40	-	-	-	40
	2628	4	F	640	640	160	-	-	-	-
	2629	2	M	40	20	20	-	-	20	40
17 March 2009 (2/1/1)	2632	3	M	40	40	20	-	-	-	-
18 March 2009 (1/1/1)	2634	2	M	40	20	40	-	-	-	20
20 March 2009 (1/1/1)	2635	3	F	40	40	20	-	-	-	40
24 March 2009 (4/3/3)	2637	3	F	80	40	40	-	-	-	40
	2638	U	F	40	40	40	-	-	-	-
	2640	2	M	80	40	20	-	-	-	20
2 April 2009 (2/1/1)	2646	3	F	40	40	20	-	-	-	-
7 April 2009 (4/2/2)	2651	U	U	20	20	20	-	-	-	80
	2652	U	U	40	40	20	-	-	-	20
8 April 2009 (2/2/2)	2654	2	M	40	40	-	-	-	-	20
	2656	2	M	40	40	40	-	-	-	20
21 April 2009 (2/1/1)	2660	3	F	80	80	40	-	-	-	20
24 April 2009 (4/1/1)	2669	U	U	20	40	20	-	-	-	40
29 April 2009 (7/5/5)	2671	2	M	40	40	80	-	-	-	-
	2672	2	F	80	40	40	-	-	-	-
	2673	2	F	80	80	40	-	-	-	-
	2674	3	F	160	80	80	-	-	-	-
	2675	3	M	80	80	40	-	-	-	-
7 May 2009 (3/1/1)	2690	4	F	40	20	-	-	-	-	-
12 May 2009 (2/1/1)	2707	3	M	20	20	20	-	-	-	40
19 May 2009 (4/1/1)	2711	2	M	80	40	40	-	-	-	-
30 June 2009 (8/1/1)	2754	2	M	320	80	40	-	-	-	-
2 July 2009 (8/2/2)	2756	3	M	80	40	40	-	-	-	-
	2757	2	F	160	80	80	-	-	-	-
16 July 2009 (2/1/1)	2781	3	F	80	80	40	-	-	-	-
30 September 2009 (1/1/1)	2809	4	M	160	80	160	-	-	-	-
20 October 2009 (12/3/3)	2829	3	F	320	160	80	-	-	-	20
	2832	3	F	160	80	40	-	-	-	-
	2833	3	F	160	80	40	-	-	-	-
11 December 2009 (8/1/1)	2894	3	M	320	80	40	-	-	-	-
7 May 2010 (3/1/1)	2940 (^§^)	3	M	640	640	320	-	-	-	-
18 May 2010 (3/2/2)	2955 (^§^)	4	M	160	80	40	-	-	-	-
	2957	U	U	160	80	40	-	-	-	20
19 May 2010 (6/1/1)	2959	3	F	160	160	160	-	-	-	-
31 May 2010 (3/1/1)	3002	3	F	160	160	80	-	-	-	-
4 June 2010 (14/2/2)	3007	3	F	640	320	320	-	-	-	-
	3019	1	F	40	20	20	-	-	-	-
18 June 2010 (9/1/1)	3044	3	M	1280	320	320	-	-	-	-
27 July 2010 (1/1/1)	3092	4	M	40	40	20	-	-	-	-
3 September 2010 (1/1/1)	3104	3	F	80	40	40	-	-	-	40
6 June 2011 (2/1/1)	3303	3	M	-	-	-	-	-	-	20
14 June 2011 (4/1/1)	3322	3	F	20	-	-	-	-	-	-
15 July 2011 (3/1/1)	3361	1	F	-	-	-	-	-	-	20
23 July 2011 (2/1/1)	3376	3	F	320	160	40	-	-	-	-
7 October 2011 (2/1/1)	3507	4	F	80	40	40	-	-	-	-
11 November 2011 (1/1/1)	3576	4	M	80	20	20	-	-	-	-
1 June 2012 (1/1/1)	3803	4	F	160	80	40	-	-	-	-
21 June 2012 (2/1/1)	3806	3	M	-	-	-	-	40	20	80
20 September 2012 (10/2/1)	3852	3	F	-	-	-	-	20	-	-
26 September 2012 (6/1/1)	3874	3	M	-	-	-	-	-	-	20
5 October 2012 (3/2/2)	3903	1	M	-	-	-	-	20	-	-
	3904	1	M	-	-	-	-	20	-	-
4 March 2013 (4/1/1)	4091	4	F	20	20	20	-	-	-	-
12 June 2014 (6/2/1)	4716	4	M	-	-	-	-	-	-	20

No., wild boar number; pos., positive; ID, identification; F, female; M, male; U, undetermined; *, Age class (in months): 0 mo. < age class 1 ≤ 6 mo.; 6 mo. < age class 2 ≤ 14 mo.; 14 mo. < age class 3 ≤ 24 mo.; age class 4 >24 mo.; U, undetermined; -, negative (<20 HI reciprocal titer); (^§^), recaptured wild boars showing seroconversions.

## Data Availability

Not applicable.

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
