# Peer review of "Long-Term Serological Investigations of Influenza A Virus in Free-Living Wild Boars (Sus scrofa) from Northern Italy (2007–2014)"

_microorganisms, 2022, doi:10.3390/microorganisms10091768_

Round 1
Reviewer 1 Report
The authors described serological investigation of Influenza A in wild board in Italy.
The study, although based on basic serological assays, is useful as it provides insight into into the circulation dynamics of IAV wild suid populations.
The authors screened samples with ELISA test and tested only positives with HI test. This approach is reasonable since sample set is large, however, I suggest giving brief overview on the specificity/sensitivity of ELISA test used and the possibility that some positives could be missed with this approach.
The overall impression is that article is very well structured, study design and methodology are described in details.
Line 170 you have written 0% prevalence and I suggest to indicate 0.6% as it was the lowest seroprevalence with ELISA as indicated in Table 1 in 2013. In 2008 no antibodies were detected at all and this can be repeated in the text.
Reviewer 2 Report
In the present study, De Marco and colleagues carried out serological investigation using sera of wild boars collected from 2007 to 2014 in northern Italy in order to understand the epidemiological role of free-living wild boar populations in the ecology of swine influenza viruses. They found that 5.5% (145/2618) of sera were positive to influenza A viruses (IAV) and 80 sera showed HI response to swine H1N1, H1N2 and H3N2 viruses circulating in domestic pigs in Europe (northern Italy). The purpose is clear, and overall, this study is interesting work and provides valuable findings on IAV seroprevalence of wild boars.
To improve the manuscript, could the authors consider some specific points as following?
1. The most fatal flaw of this manuscript was that the number of reference papers shift after duplication of reference 1. The authors had to take careful steps.
2. In abstract, the proportion (58%, 80/138) of HI positive sera against the seven viruses used in this study should be described after the seropositivity to influenza A viruses by ELISA.
3. Across the Results section, the texts were complicated and hard for me to understand. Could you please give extra consideration to describe concisely?
4. Table 1 can be refined. At first, it is confusing that “the percentage (%)” and “(positive number/sample number)” were arranged horizontally in a line and vertically in another line. What is “^” at upper right of 1 of Age class? Why is “2007” described in bold font? What is a line between 2007 and 2008?
5. Lines 166-168, the sentence “Twenty-seven wild boars~” is almost repetition of MM section (107-109). Only essential point should be shown.
6. Lines 172-174, the description about significant difference in seroprevalence values between years was difficult to understand. If possible, could you show these significant differences in figure or table?
7. Lines 176-181, the description about significant difference in seroprevalence values between ages was also difficult to understand. For example, the number of positive wild boars in each age class can be shown in Table 1.
8. Lines 184-185, why did IAV positive sera decrease from 145 to 138? Could you show the reason in the manuscript why 7 IAV positive sera were not available for HI test. In Table 2, available wild boar sera tested positive for IAV was 141. In supplementary Table, 148 sera of wild boars were screened as positive for IAV. Inconsistency of the IAV-positive numbers could fill the audiences with confusion.
9. Table 2 can be refined. The name of viruses was not aligned properly in the second and third pages.
10. Lines 298-299, Figure 1A is typo of Figure A1.
11. Lines 321-327, the number of IAV-positive samples in these sentences were inconsistent to Table 1. The numbers of IAV positive sera were 51 in 2009, 9 in 2011, and 20 in 2012 in Table 1, whereas 47 in 2009, 8 in 2011, and 18 in 2012 in the main text.
Round 2
Reviewer 2 Report
The revised manuscript by De Marco et al. improved and become easier to understand. However, I am a little confused by handling data from 3 recaptured wild boars. Some results count recaptured wild boars, but others do not count, making me confusing.
I am afraid that the following sentences may include description contrary to fact. Because I could be wrong, could you please check these points?
1. At lines 196-197 in the revised manuscript, the authors describe that HI positive results were obtained from 80 out of 138 available IAV positive sera. Among 145 ELISA(+) sera, seven sera were not available for HI test because of small volume. Alternatively, ELISA(+) sera from 3 recaptured wild boars were available. Accordingly, the authors obtained 80 HI(+) sera. Therefore, I think “80 out of 141” is true and it could be easy to understand for the audiences.
2. At lines 203-204, the authors also describe that 58% of sera tested IAV positive were found to be HI positive for at least one sIAV strain. "58%" could be calculated from 80/138. Actually, 56.7% (=80/141) is appropriate. Similarly, in abstract at line 28-29, “58% (80/138)” should be “56.7% (80/141)”.
3. At lines 370-372 in caption of Table S1, the authors describe that HI titers obtained from “138/148” sera of wild boars screened as positive for IAV by ELISA. Denominator number of 148 includes three recaptured wild boars. If so, “141/148” is likely correct.
Minor comments:
At line 162, "1233 females, 1241 males and 144 unsexed animals". In Table 1, the numbers of male and undetermined were revised to 1239 and 146. Either is wrong.
At line 335, “0% (0/4” is typo of “0% (0/4)”.
At line 351, “Figure 1A” should be “Figure A1”.
